# A Detailed Comparative Analysis of the Structural Stability and Electron-Phonon Properties of ZrO_2_: Mechanisms of Water Adsorption on t-ZrO_2_ (101) and t-YSZ (101) Surfaces

**DOI:** 10.3390/nano13192657

**Published:** 2023-09-27

**Authors:** Dilshod D. Nematov, Amondulloi S. Burhonzoda, Kholmirzo T. Kholmurodov, Andriy I. Lyubchyk, Sergiy I. Lyubchyk

**Affiliations:** 1Osimi Tajik Technical University, Dushanbe 734042, Tajikistan; 2S.U. Umarov Physical-Technical Institute of NAST, Dushanbe 734042, Tajikistan; 3Joint Institute for Nuclear Research, 141980 Dubna, Russia; 4Dubna State University, 141980 Dubna, Russia; 5DeepTechLab, Lusophone University, 1749-024 Lisbon, Portugal

**Keywords:** zirconia, stability, yttrium-stabilized ZrO_2_ (YSZ), phase transition, doped-induced phase transition, Fermi level shift, oxygen vacancy, enthalpy and entropy, water adsorption on the surface

## Abstract

In this study, we considered the structural stability, electronic properties, and phonon dispersion of the cubic (c-ZrO_2_), tetragonal (t-ZrO_2_), and monoclinic (m-ZrO_2_) phases of ZrO_2_. We found that the monoclinic phase of zirconium dioxide is the most stable among the three phases in terms of total energy, lowest enthalpy, highest entropy, and other thermodynamic properties. The smallest negative modes were found for m-ZrO_2_. Our analysis of the electronic properties showed that during the m–t phase transformation of ZrO_2_, the Fermi level first shifts by 0.125 eV toward higher energies, and then decreases by 0.08 eV in the t–c cross-section. The band gaps for c-ZrO_2_, t-ZrO_2_, and m-ZrO_2_ are 5.140 eV, 5.898 eV, and 5.288 eV, respectively. Calculations based on the analysis of the influence of doping 3.23, 6.67, 10.35, and 16.15 mol. %Y_2_O_3_ onto the m-ZrO_2_ structure showed that the enthalpy of m-YSZ decreases linearly, which accompanies the further stabilization of monoclinic ZrO_2_ and an increase in its defectiveness. A doping-induced and concentration-dependent phase transition in ZrO_2_ under the influence of Y_2_O_3_ was discovered, due to which the position of the Fermi level changes and the energy gap decreases. It has been established that the main contribution to the formation of the conduction band is made by the p-states of electrons, not only for pure systems, but also those doped with Y_2_O_3_. The t-ZrO_2_ (101) and t-YSZ (101) surface models were selected as optimal surfaces for water adsorption based on a comparison of their surface energies. An analysis of the mechanism of water adsorption on the surface of t-ZrO_2_ (101) and t-YSZ (101) showed that H_2_O on unstabilized t-ZrO_2_ (101) is adsorbed dissociatively with an energy of −1.22 eV, as well as by the method of molecular chemisorption with an energy of −0.69 eV and the formation of a hydrogen bond with a bond length of 1.01 Å. In the case of t-YSZ (101), water is molecularly adsorbed onto the surface with an energy of −1.84 eV. Dissociative adsorption of water occurs at an energy of −1.23 eV, near the yttrium atom. The results show that ab initio approaches are able to describe the mechanism of doping-induced phase transitions in (ZrO_2_+Y_2_O_3_)-like systems, based on which it can be assumed that DFT calculations can also flawlessly evaluate other physical and chemical properties of YSZ, which have not yet been studied quantum chemical research. The obtained results complement the database of research works carried out in the field of the application of biocompatible zirconium dioxide crystals and ceramics in green energy generation, and can be used in designing humidity-to-electricity converters and in creating solid oxide fuel cells based on ZrO_2_.

## 1. Introduction

With the ongoing threat of the energy crisis and global warming caused by the increase in the use of fossil fuels, the search for sustainable and environmentally friendly sources of energy is one of the most pressing challenges facing human civilization today [1,2]. Fossil fuels’ continued use worldwide threatens our energy supply and significantly burdens the environment. Research on the use of sustainable green energy represents one of the ways to mitigate the growing threat of global environmental problems and the energy crisis, which is very intense and active worldwide. Solar panels and wind turbines have become familiar to us. However, new advances in nanotechnology and materials science make it possible to collect energy from other sources, and will help to implement Nikola Tesla’s idea of “Getting an electric current from the air”. Recently, scientists and engineers have been developing innovative devices for converting humidity into electricity, which will expand the range of known renewable energy sources. These use galvanic converters that convert air humidity into electricity. Such devices can collect electricity from atmospheric humidity and supply an electrical current, similar to how solar panels capture sunlight and generate electricity.

Moisture, which is ubiquitous on Earth (approximately 71% of the Earth’s surface is covered by water), contains a large reservoir of low-potential energy in the form of gaseous water molecules and water droplets. It has been found that a number of functional nanomaterials, such as TiO_2_, CaSiO_3_, ZrO_2_, SnO_2_, and Al_2_O_3_, as well as biofibers and carbon materials, can generate electricity directly when interacting with moisture [2]. This presents the possibility of generating electrical energy from atmospheric moisture, allowing the creation of self-powered devices. While this technology is still evolving, there are already some strategies for improving the energy conversion efficiency and power output of these devices. 

Zirconium ceramics have been extensively studied in recent years because of their excellent electrical, optical, and mechanical properties. They are also biocompatible and have a wide range of biomedical applications. Tetragonal phase yttria-stabilized zirconia (Y-TZP) has been used in various medical applications since the 1980s, particularly for dental crowns [2]. In addition, bulk materials and nanocomposites based on ZrO_2_ are used in electrochemical cells because of their high oxide ion conductivity and catalytic activity, low thermal conductivity, mechanical/chemical stability, and compatibility with electrolytes, which make them structurally advantageous [3,4].

One of the most remarkable properties of ceramics based on zirconia is the presence of three crystalline forms with different properties [5,6,7,8,9], namely a stable monoclinic structure (mineral baddeleyite; m-ZrO_2_), a metastable tetragonal structure (medium temperature), and an unstable cubic structure (high temperature). High-pressure-induced zirconium phases in the form of brookite (orthorhombic-I) and cotunnite (orthorhombic-II) are also known [10,11].

Pure zirconium dioxide undergoes a phase transformation from monoclinic to tetragonal (at about 1173 °C) and then to cubic (at about 2370 °C), accompanied by a change in volume, and accordingly, strength [12,13,14]. For the application of zirconia in advanced zirconia ion-conducting ceramic devices, it is important that the stabilized material has an adequate level of conductivity and that it has the desired mechanical–chemical stability in both oxidizing and reducing atmospheres. Obtaining a stable material from zirconia is difficult due to a noticeable change in volume during the phase transition. Stabilization of zirconium dioxide is achieved by replacing some Zr^4+^ ions with larger ions in the crystal lattice [15,16,17]. For example, numerous studies have shown that doping with polyvalent oxides, including certain concentrations of yttrium oxide, stabilizes the high-temperature cubic and tetragonal phases of ZrO_2_ at room temperature. This also leads to an increase in the concentration of oxygen vacancies and in oxygen-ion conductivity, which makes it possible to use stabilized ZrO_2_ as an electrolyte in fuel cells [17]. The ionic conductivity of ZrO_2_ strongly depends on the phase modification and the content of stabilizing additives in the system, as shown in the phase diagram given in [18].

Many technological applications of zirconia (pure ZrO_2_ or its stabilized alloys) are directly related to its interaction with water. Examples are internal steam reforming in solid oxide fuel cells [19], catalysis [20], gas sensors [21], or its use as a biocompatible material [22]. ZrO_2_ surfaces have also been proposed as suitable materials for hydrogen storage [21,22,23]. However, little is known about the interaction of water with ZrO_2_ surfaces at a fundamental level, which is mainly due to the lack of suitable samples. This is quite different for other oxide substrates [23,24,25]. Water is weakly adsorbed by many defect-free oxide surfaces in an ultra-high vacuum, then stripped at a temperature below room temperature. Usually, at 160–250 K [26], water can bind more strongly to surfaces with defects, as was shown for rutile (TiO_2_ (110)) [27]. In these cases, H_2_O dissociates into an OH group, which fills the oxygen vacancy, and into a hydrogen atom, which binds to surface oxygen and forms a second OH group. These OH groups are stable for up to 490 K on TiO_2_ [28]. On a defect-free surface oxides (e.g., a-Cr_2_O_3_ (001) [29], a-Fe_2_O_3_ (012) [30], and oxides of alkaline earth metals, including Ca_3_Ru_2_O_7_ (001) [31]), water can be strongly bound if the end of the surface includes highly active cations. Then, it can easily dissociate. On the surfaces of RuO_2_ (110), PdO (101), and Fe_3_O_4_ (001), water binds coordinatively unsaturated cations and partially dissociated forms of the structure stabilized by hydrogen bonds [32,33,34]. High-enthalpy adsorption of low-H_2_O powder materials (≥2 eV on monoclinic and ≈1.5 eV on tetragonal ZrO_2_) has been reported to decrease liquid–water binding (0.45 eV) at coverages of approximately 2–4 H_2_O/nm^2^ [35]. In another study, Droshkevich et al. [36] reported on the chemo-electronic conversion of water adsorption energy into electricity on the surface of zirconium dioxide nanopowders that were synthesized at sizes of 7.5 nm, when doped with 3 mol. %Y_2_O_3_. 

However, despite numerous studies in this area, water adsorption on ZrO_2_ surfaces has not been studied in detail, and only a few reports on H_2_O adsorption can be found in the literature. In particular, H_2_O adsorption on well-defined monoclinic surfaces of zirconia (m-ZrO_2_ (101) and m-ZrO_2_ (101) and its doped structures) has not been studied. For example, it is especially difficult to experimentally study pure ZrO_2_ single crystals grown from a melt; they exhibit phase transformations upon cooling; therefore, their doped structures (e.g., YSZ) are usually investigated. However, the surface chemistry of YSZ is more complex than pure ZrO_2_, as shown for CO and CO_2_ adsorption [37]. In another work, Kobayashi et al. [38] found that YSZ slowly decomposed at about 250 °C due to the t–m transformation. In a humid atmosphere, this t–m transformation is accompanied by microcracks and a loss in material strength. This discovery cooled the excitement caused by the discovery of PPT in zirconia-based ceramics. This t–m transformation due to the presence of water or a humid environment in zirconia-based ceramic materials has been termed low-temperature degradation, or aging of ZrO_2_ crystals. This topic has been researched extensively over the past couple of decades, including many hypotheses and discussions. The most reliable hypothesis about YSZ is based on filling oxygen vacancies present in the matrix to maintain a stable t-YSZ phase. Thus, the filling of these O vacancies with water radicals, either O_2_ or OH, destabilizes the t-YSZ phase. However, the YSZ stabilization mechanism has not been fully studied, and is still the subject of numerous discussions. Therefore, the theoretical study and modeling of water adsorption on these surfaces is necessary as a starting point for a good understanding of ongoing processes and phenomena from a fundamental point of view. On the other hand, aspects of the shift in the Fermi level after doping with yttrium oxide in ZrO_2_, as well as when it is under the influence of water adsorption, are still not clear due to the difficulty of their detection.

For these reasons, to obtain detailed information on the process of the adsorption of water molecules onto ZrO_2_ and YSZ surfaces, as well as the effect of doping on their electronic and structural properties, we conducted quantum chemical calculations within the framework of density functional theory (DFT).

## 2. Modeling Details

First-principles (ab initio) methods within the framework of DFT are successfully used in modern materials science regarding condensed matter physics [39,40,41,42,43,44,45,46,47,48].

We conducted ab initio quantum chemical calculations on the basis of density functional theory [49]. All three phases of ZrO_2_ (Figure 1a–c) were first relaxed using generalized gradient approximation (GGA) functionals (PBE) [50] and strictly bounded normalized potentials (SCAN) [51]. To obtain the most accurate value of the ground state energy, the total energy was calculated within the framework of the GGA exchange–correlation potential and SCAN were used to correctly estimate the lattice parameters. The calculations were carried out using the Vienna ab initio Simulation Package (VASP 6.3.2) [52]. We found a stable ZrO_2_ phase by comparing the total energy in the unit cell. For stabilization at room temperature, a 2 × 2 × 2 supercell was created to simulate the effect of 3.23, 6.67, 10.34, and 16.15 mol. %Y_2_O_3_ on the stability of ZrO_2,_ as well as to evaluate the influence of Y_2_O_3_ doping on the position of the Fermi level. We performed an orbital analysis by summing the contributions of the individual atomic species in the unit cell and showing the contributions of the main atoms at the meeting point of the valence and conduction bands. Vacancies were taken into account by removing one O atom with each subsequent substitution of 2 Y^3+^ ions to the Zr^4+^ position. The atomic orbitals of H (1^s^), O (2^s^, 2^p^), Zr (4^d^, 5^s^), and Y (4^s^, 4^p^, 4^d^, 5^s^) were considered valence electrons, while the remaining electrons were considered nuclear electrons and remained frozen. The PAW method was used to describe the interactions between valence electrons and electrons in the nucleus. The kinetic energy cutoff was fixed at 600 eV, and all calculations were carried out while taking spin-polarized effects into account. 

Next, ab initio calculations were carried out to study the mechanism of the adsorption of a water molecule on the surface of ZrO_2_ and ZrO_2_ by stabilized Y_2_O_3_, where we found the adsorption energies of a water molecule on ZrO_2_ and YSZ surfaces. We also conducted an orbital analysis and estimated the shift in the Fermi level.

For such specific problems, the choice of the adsorbed surface is very important. To obtain results consistent with the experiment, we had to accurately select a suitable surface with the lowest density of broken surface bonds and electrostatic repulsion of neighboring layers, while considering the thermodynamic stability of the surface. The higher the surface energy, the more thermodynamically unstable it is [53] and the more difficult it is to create the corresponding surface, because surface energy is closely related to the number of atoms in the surface structure and the depth of the vacuum layer.

To select a suitable optimal surface for water adsorption and study the water’s behavior on this surface, we calculated the surface energy (*σ*) using Equation (1) [54]:(1)σ=12Eslab−NnEbulk S
where *S* is the total surface area of the plate; $E_{slab}$ is the total plate energy; $E_{bulk\;}$ is the total energy of an optimized bulk structure; *N* and *n* represent the total numbers of atoms in the surface structure and unit cell, respectively; and 2 represents the two surfaces of the calculated structure in the direction of the z-axis.

Models of the crystal wafer surface were constructed based on an extended 2 × 2 supercell with a vacuum space of 35 Å along the z-direction to minimize the interaction of neighboring layers. Taking into account the accuracy and calculation time, the lower layers of the surface plate were frozen, and the upper part was allowed to relax. Monkhorst–Pack grids with a 3 × 3 × 1 k-point grid were used to sample the reciprocal space for the 2 × 2 plate calculations. Each molecule in the gas phase was placed in a large box (11 × 13 × 10 Å^3^) to avoid side interactions. 

Single H_2_O molecules were initially located at a height of 2.5 Å above the chosen surface; different orientations, the relaxing of H_2_O molecules, and the upper layers of the plate for each initial adsorption site were compared. For each molecule, we tested four initial adsorption centers (above the Zr atom, above the terminal oxygen Ou (top) or Od (bottom), and in the center above the Zr position (see Figure 1d)). We also investigated various initial adsorption sites for the YSZ surface model—namely, above the Zr atom, the extreme oxygens, Ou (top) and Od (bottom), in the oxygen vacancy position, the yttrium atom, and the Ou–Od–Zr center (see Figure 1e)—to determine the most favorable adsorption sites leading to stable configurations. We did not study the nonequivalent initial adsorption sites in detail. 

The adsorption energy ($E_{ads\;}$) was calculated as the difference between the energy of the plate with adsorbed water ($E_{H2O/surface}$) and the sum of the energies of the surface ($E_{surface}$) and H_2_O molecules ($E_{H2O}$), according to the following equation: (2)$$E_{ads\;}=E_{H2O/surface}-(E_{surface}+E_{H2O})$$

To take into account long-range uncoupled interactions, we considered Van der Waals effects as the difference between the calculated Van der Waals energy of a plate with adsorbed H_2_O molecules ($E_{H2O\;/surface}^{vdW}$) and the sum of the calculated Van der Waals energies of the surface ($E_{surface}^{vdW}$) and H_2_O molecules ($E_{H2O\;}^{vdW}$):(3)$$E_{ads\;}^{vdW}=E_{H2O\;/surface}^{vdW}-(E_{surface}^{vdW}+E_{H2O\;}^{vdW})$$
where the interaction energy vdW is taken into account by the Leonard–Jones potential.

## 3. Results and Discussion

### 3.1. Structural Stability and Electron-Phonon Properties of ZrO_2_


In the first stage of modeling, the structural energy relaxation of pure ZrO_2_ phases was carried out using the VASP package. To find the optimal cutoff energy for the ENCUT plane-wave basis functions and the corresponding number of k-points in the Brillouin zone, we tested the convergence of the total unit cell energy as a function of ENCUT and KPOINTS. 

Convergence tests to select an appropriate k-point grid were first performed for all three ZrO_2_ phases with an initial value of ENCUT = 1.3 × ENMAX. The smallest grid of 4 × 4 × 4 k-points with the Monkhorst–Pack scheme was optimal for the geometric relaxation of all studied ZrO_2_ phases. However, when calculating the electronic structures of these compounds, the number of k-points was at least doubled to obtain a better density of states (DOSs). 

Similar tests were carried out to establish the cutoff energy. They showed that 600 eV was suitable for the calculations, and further increasing this energy increased the cost of calculations without improving their accuracy. Therefore, all further calculations were conducted at ENCUT = 600 eV.

According to the results given in Table 1, lattice distortion during the transition between the high- and low-temperature phases causes a displacement of O ions in the c-direction by the value of dz, expressed in relative units. As a result of distortion in the tetragonal phase, all Zr–O bonds will become nonequivalent.

Table 2 compares the total unit cell energies calculated using the GGA method for the monoclinic, tetragonal, and cubic phases of ZrO_2_. Among all systems, m-ZrO_2_ is the most stable phase with the lowest energy. That is, in terms of field energy at low temperatures, the stable phase is monoclinic with the space group P21/c.

As shown in Table 3, the SCAN functionality describes the geometry better than the standard GGA-PBE. However, the available data also show that GGA and SCAN describe the energy difference between the monoclinic and tetragonal phases of ZrO_2_ almost identically. Since the SCAN exchange–correlation functional describes the structural properties well, we decided to use this functional in the future when describing the geometry of other systems.

Furthermore, we calculated the thermodynamic properties and phonon spectra of the ZrO_2_ phase using the Phonopy code in the VASP package for a more detailed discussion and substantiation of the ZrO_2_ monoclinic phase’s structural stability. Figure 2 shows the change in the entropy of unit cells of the ZrO_2_ phase as a function of temperature. 

Figure 2 shows that with the transition from the monoclinic phase to the tetragonal and cubic phases, the entropy of these compounds decreases, which corresponds to the criterion of the inverse dependence of enthalpy or direct dependence of entropy on the stability of solid systems [63]. Thus, the monoclinic phase is the most stable, with the highest entropy among the three ZrO_2_ phases. This pattern can be clearly observed after analyzing the pattern of phonon frequencies of the three ZrO_2_ phases (Figure 3a–c), from which it is clearly seen that the monoclinic phase has smaller negative modes than the other two phases.

Figure 4a–c shows the temperature dependence of the free energy, entropy, and heat capacity of a 12-atom supercell for m-ZrO_2_, t-ZrO_2_, and с-ZrO_2_.

Figure 5a–c presents the results of the calculations of the density of phonon states, which indicate that during the transition from the monoclinic phase to the tetragonal and cubic phases, the density of electronic states increases. These results also agree with those shown in Figure 3, confirming that the monoclinic phase is the most stable among the ZrO_2_ phases. The energy/volume chart presented by Teter et al. also supports these results [64]. Therefore, for further stabilization by doping with Y_2_O_3_, it is reasonable to choose a monoclinic phase. 

Next, using the well-optimized structures of the three phases of ZrO_2_, we performed calculations to study their electronic properties. We found the band gaps of these systems (Table 3) using the GGA and SCAN functionals, as well as the HSE06 hybrid functional, analyzed their orbital structure, and modeled the change in the position of the Fermi level. 

As seen in Table 3, the GGA and SCAN functionals showed a rather small band gap compared to the HSE06 hybrid functional [65], the results of which were much closer to the experimentally determined band gap. On the other hand, the standard SCAN and GGA functionals greatly underestimate the band gap. Given the suitability of HSE06 for estimating the band gap energy, we further used this hybrid functional to describe all the problems associated with the electronic properties of the systems under study.

In the next step, we calculated the density of available electronic states at the Fermi level for ZrO_2_ structures relaxed using the HSE06 functional (Figure 6), which is crucial for interpreting the electronic properties of ZrO_2_ and the transport characteristics of electronic devices. 

Figure 7 shows the partial density of states for the three ZrO_2_ phases. The contribution of Zr atoms to the formation of the valence band is greater than that of oxygen O, and the contribution of O atoms to the formation of the conduction band is greater than that of Zr atoms.

According to Figure 6, the density of electronic states for c-ZrO_2_ is somewhat overestimated compared to the other phases. In addition, secondary energy gaps are observed in the energy diagram of the tetragonal and cubic phases. Furthermore, this gap increases during the transition from the tetragonal phase to the cubic phase. 

Next, we determined the position of the Fermi level in ZrO_2_ crystals and its shift during their phase transformation. As seen in Figure 8, if we assign the position of the Fermi level (maximum of the valence band) for the monoclinic phase as a reference point, this level first shifts by 0.125 eV towards higher energies during the m–t phase transformation of ZrO_2_ (towards the valence band). Then, in the t–c section, it decreases by 0.08 eV. This is also observed in the band stacking results of the orbital analysis, which are shown in Figure 9 for the three phases of ZrO_2_.

During the transition from the monoclinic to the tetragonal and cubic phases, the contribution of the p-orbitals becomes more significant in the conduction band. The s-orbitals make a small contribution, while the d state shows a different trend. This behavior may be associated with a change in the crystal field and covalence of ZrO_2_ during the phase transformation.

### 3.2. Structural and Energy Properties of m-ZrO_2_ Doped with Y_2_O_3_: The Electronic Properties of YSZ

We created supercells of 96 atoms with a size of 2 × 2 × 2 to simulate the effect of Y_2_O_3_ on the stability and electronic properties of the most stable (monoclinic) ZrO_2_ phase. To dope ZrO_2_ with yttrium, it was necessary to replace some formula units of ZrO_2_ with Y_2_O_3_ in a 2 × 2 × 2 supercell, with each replacement creating one oxygen vacancy. A schematic description of the generation of YSZ structures is provided below:$${xZrO}_{2\;}+kY_{2}O_{3}\;\to \;{Zr}_{x}Y_{2k}O_{2x+3k}+V_{o_{k}}\;\%Y_{2}O_{3}=\frac{k}{x+k}\times 100\%,$$
which can be considered the union of ***x*** ZrO_2_ with the *k*Y_2_O_3_ formula units located on the initial lattice of the *x + k*ZrO_2_ units, leading to the formation of m oxygen defects. Based on this, we determined that the percentage of vacancies is equal to the percentage of yttrium units in the final structure. Thus, starting with a pure 96-atom ZrO_2_ supercell, we mainly focused on four different concentrations of Y_2_O_3_ in our calculations (Table 4).

After the final preparation of the YSZ structures, we performed geometric optimization and doping relaxation of the Y_2_O_3_ supercell using the GGA and SCAN potentials. Figure 10 shows a diagram of the dependence of the change in the enthalpy of formation of YSZ on the concentration of Y_2_O_3_, calculated using Formula (4):(4)$$\Delta H=\frac{E_{YSZ}-[xE_{ZrO_{2}}+kE_{Y_{2}O_{3}}]}{x+k},$$ from which it is clearly seen that doping with Y_2_O_3_ reduces the enthalpy and leads to the stabilization of zirconium dioxide. The empirical formula obtained using the least squares method states that the enthalpy of formation energy decreases linearly according to the law ΔН = −1.0407x + 63.532, where x is the concentration of Y_2_O_3_ in YSZ.

Thus, the number of oxygen vacancies in YSZ increases with the increase in the Y_2_O_3_ concentration, and the growth of these O vacancies is considered a stabilizing mechanism of the monoclinic zirconium phase, as indicated by a decrease in the enthalpy of formation. 

Table 5 shows the geometric parameters of the ZrO_2_ and YSZ supercells at various Y_2_O_3_ concentrations after thorough relaxation using the SCAN functional. Table 6 presents the numerical values of the enthalpy of formation energies.

After obtaining the optimized structures, the energy of formation (*E_f_*) for ZrO_2_ and YSZ, as well as the energy of formation of vacancies (*E_df_*) for YSZ were calculated as follows:$$E_{f}=E_{tot}-\sum_{x}E_{tot}(x)\;E_{df}=E_{tot}^{{Zr}_{32-x}Y_{x}O_{64-\delta }}-E_{tot}^{{Zr}_{32}O_{64}}+\delta \times E_{tot}^{O},$$
where $E_{tot}$ is the total energy of the system, $E_{tot}(x)$ is the total energy of individual components, and δ is the number of vacancies (defects) in the crystal. Table 6 presents the calculated values of $E_{f}$ and $E_{df}$ for each atom. 

Figure 11 shows the nature of the changes in $E_{f}$ and $E_{df}$ with the yttrium oxide concentration, from which the regularity of their linear decrease is clearly visible.

Next, calculations were performed to study the electronic structure of Y_2_O_3_-stabilized ZrO_2_ supercells to reveal the effect of doping on the density of states, the behavior of the Fermi energy, and the orbital components. Figure 12 shows plots of changes in the density of electronic states of YSZ for all doping concentrations of Y_2_O_3_. 

As seen in Figure 12, new energy states do not appear in the TDOS patterns after doping with Y_2_O_3_ due to the introduction of defects. In other words, there are no noticeable changes except for a decrease in the band gap, which can be explained by orbital analysis (Figure 13) and Fermi level mixing estimates (Figure 14). The band gaps are 4.71 eV, 4.92 eV, 4.75 eV, and 4.72 eV, respectively, for ZrO_2_ doped with 3.23, 6.67, 10.34, and 16.15 mol. %Y_2_O_3_.

In Figure 13, after doping 3.23 mol. %Y_2_O_3_ into pure m-ZrO_2_, the Fermi level drops by 0.067 eV, and then shifts by 0.007 eV toward the conduction band upon doping with 6.67 mol. %Y_2_O_3_. At a doping concentration of 10.34 mol. %Y_2_O_3_, it still increases by 0.01 eV, which is 0.017 eV more than in the case of 3.23 mol. %Y_2_O_3_. However, after doping with 16.15 mol. %Y_2_O_3_, it drops to 0.012 eV. The PDOS diagram also interprets the stepped conduction band pattern in terms of the s-, p-, and d-orbital contributions. Understanding these features makes it possible to tune the Fermi energies in the band structure to solve the most important problems of materials science and instrumentation.

The problems of studying the influence of yttrium oxide doping on the properties and stability of tetragonal and cubic zirconia remains the subject of future research.

### 3.3. Water Adsorption on ZrO_2_ and YSZ Surfaces

As previously mentioned, to model the water adsorption mechanism on the corresponding surface correctly, the most important point is the choice of the surface with the lowest surface energy. To select the optimal adsorbing surface, we calculated the surface energy (σ) for several different surface models according to Equation (1) after their geometric relaxation. Table 7 shows the calculated values of the surface energies of ZrO_2_.

As shown in Table 7, the most stable surfaces can be obtained from the tetragonal and monoclinic phases, namely t-ZrO_2_ (101) and m-ZrO_2_ (111). The results obtained agree qualitatively with Maliki et al. [66], who reported that the most stable surface can be obtained from t-ZrO_2_ (101). As for comparing the results to experimental data, there are no reported data in the literature because the surface energies of solid metal oxides are difficult to measure experimentally. In total, measurements of the surface energies of some types of zirconium dioxide surfaces using multiphase balancing at high temperatures has been reported [67]. Based on the results obtained, we chose the t-ZrO_2_ (101) surface for this study, as the most stable surface for water molecule adsorption.

After the final surface preparation, single H_2_O molecules were initially located at a height of 2.5 Å above the selected surface with different orientations, which is greater than the bond distance between Zr and O (2.12 Å) in the solid state. The structures were then optimized by freezing the bottom layers of the wafer (Figure 15a). 

The optimized structure of the H_2_O + t-ZrO_2_ (101) system is shown in Figure 15b, which shows that the H_2_O molecule is dissociatively adsorbed with an energy of −1.221 eV, even in the most favorable region (where the system has the minimum energy of the stable configuration). Korhonen et al. [68] also observed dissociative adsorption of water on ZrO_2_, where it was experimentally and theoretically proven that water dissociates on the surface of m-ZrO_2_ at a low coverage. The adsorption energy we calculated on t-ZrO_2_ (101) for [H+OH]-ZrO_2_ (101) is similar to their results for monoclinic (111) and (101) surfaces with an energy of −1.20 eV. We also found that water is adsorbed on this surface via molecular chemisorption, in which the water’s oxygen coordinates the surface cation, and a slight elongation of one O–H water bond (1.13 Å) occurs in the form of hydrogen bonding of water with the surface oxygen ion (Figure 15c). In this case, the adsorption energy is 0.69 eV, and the distance between the oxygen of the water molecule and the surface zirconium atom is 2.205 Å. In this case, the proton (H) in the water molecule and oxygen from the surface of the plane form a hydrogen bond with a bond length of 1.01 Å. 

To study the mechanism of water adsorption on the surface of t-YSZ, we replaced two Zr (from the uppermost and subsurface O–Zr–O trilayers) by Y with the removal of one oxygen from the third nearest atomic layer to the Y atoms to obtain a surface similar to t-YSZ (101). Our results showed that the water molecule is molecularly adsorbed and also dissociated on the t-YSZ (101) surface. Molecular adsorption of water in the optimal configuration occurs at an energy of −1.84 eV, and the bond length of water with the t-YSZ (101) surface increases to 2.73 Å (Figure 16a). In this case, the O–H distance in the water molecules will remain unchanged. 

The dissociative adsorption of water was accompanied by the movement of oxygen in the area of the plate vacancies, leading to strong adsorption of −1.23 eV, and blocking surface areas for oxygen activation. In both cases, H_2_O is adsorbed near the yttrium atom (Figure 16b).

Unlike water adsorption on t-ZrO_2_ (101), H_2_O is more stably adsorbed on t-YSZ (101) since the adsorption energy of H_2_O–YSZ (101) is more favorable than (H+OH)–YSZ (101).

Doping with Y_2_O_3_ stabilizes t-ZrO_2_ (101) and is accompanied by large relaxations of O atoms. Calculations based on the GGA functional greatly underestimate the band gap of the system (3.24 eV for H_2_O–ZrO_2_ (101) and 3.21 eV for H_2_O–YSZ (101)). However, despite the presence of the Oth vacancy, the average gap energy states did not appear in the t-YSZ band diagram, as observed in the systems under study. A comparative analysis of the H_2_O–ZrO_2_ (101) and H_2_O–YSZ (101) systems’ electronic structures indicate that the H_2_O interaction practically does not change the electronic configuration of the system (except for an increase in the density of state) during the transition of the system to being modified by Y impurities (Figure 17). However, water molecules are predominantly prone to molecular adsorption on the t-YSZ (101) surface, whereas they are more often dissociatively adsorbed on t-ZrO_2_ (101). Table 8 lists some key data obtained by modeling water adsorption on the t-ZrO_2_ (101) and t-YSZ surfaces. 

In such studies, it is also important to take into account the hydrophilic nature of ZrO_2_. Studies show that in addition to physically adsorbed water, the substrate surface also contains terminal, bi-bridging, and triple bridging OH groups, which are actively involved in the surface reaction [69,70,71,72,73,74,75,76,77,78,79,80,81,82,83,84,85]. Surface hydroxyl groups and H_2_O adsorbed on the surface can partially block active sites (lattice oxygen ions on the surface) of YSZ oxidation. Figure 18a shows the surface configuration model for fully hydroxylated t-YSZ (101). The results show that the OH groups form strong bonds on the surface. Figure 18b shows the adsorption structure of a single water molecule on a fully hydroxylated YSZ surface. 

When water is adsorbed on a hydroxylated surface, two strong hydrogen bonds are formed at distances of 1.56 and 1.63 Å from each other. In this case, water is adsorbed with an adsorption energy of 0.34 eV. It can be seen that neither the repulsive forces of oxygen and hydrogen atoms in a water molecule or OH atoms on a completely hydroxylated surface prevent the adsorption of an H_2_O molecule on t-YSZ (101). The adsorption model of a single water molecule and other similar systems will help us study more complex models in detail in the future, including the multilayer hydration structure of the interface (Figure 18). Although this model requires large computational power for DFT calculations, it can be assumed that in the layer closest to the surface (hydroxyl hydration layer), most of the water molecules can be adsorbed dissociatively. Furthermore, due to hydrogen bonds, H_2_O molecules will continue to be adsorbed and be regularly located on the hydroxylated surface, forming primary and secondary hydrated layers. The regular arrangement of H_2_O molecules in the outer layer can be considered a transition layer, and the hydration structure of the first three H_2_O layers located near the surface can be considered a group of water molecules that can both be stably adsorbed and exist on the m-NSC surface (101). However, a detailed study of the complete model of t-YSZ (101) surface hydration remains the subject of future research. 

## 4. Conclusions

We investigated the stability, electronic properties, and dispersion of phonons in the three phases of ZrO_2_ using quantum chemical calculations. The stable phase is defined in terms of the total energy, enthalpy, entropy, and band structure of phonons. We established that during the m–t phase transformation of ZrO_2_, the Fermi level first shifts by 0.125 eV toward higher energies, then decreases by 0.08 eV in the t–c region. An analysis of the influence of doping 3.23, 6.67, 10.35, and 16.15 mol %Y_2_O_3_ on the m-ZrO_2_ structure showed that the m-YSZ enthalpy decreases linearly, which accompanies further stabilization of monoclinic ZrO_2_. An analysis of the mechanism of water adsorption on the surfaces of t-ZrO_2_ (101) and t-YSZ (101) showed that H_2_O on unstabilized t-ZrO_2_ (101) was adsorbed dissociatively with an energy of −1.22 eV, as well as by molecular chemisorption with an energy of −0.69 eV and the formation of a hydrogen bond with a bond length of 1.01 Å. In the case of t-YSZ (101), water was molecularly adsorbed onto the surface with an energy of −1.84 eV. Dissociative adsorption of water occurs at an energy of −1.23 eV, near the yttrium atom. Thus, with an increase in Y_2_O_3_ concentration, the number of oxygen vacancies in YSZ increases. The growth of these O vacancies is considered a stabilizing mechanism of the monoclinic zirconium phase, as indicated by a decrease in the enthalpy. These oxygen vacancies also give YSZ a high ionic conductivity, making it suitable for use in full solid oxide cells. This study will help build more accurate calculation models for other types of surfaces like YSZ by characterizing their structural and electronic properties.

## Figures and Tables

**Figure 1 nanomaterials-13-02657-f001:**
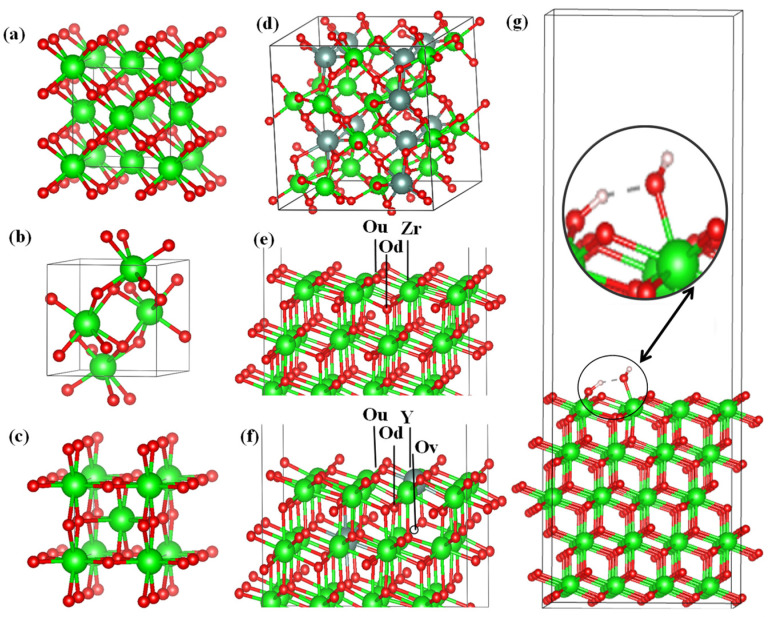
Optimized cells of the (**a**) cubic, (**b**) monoclinic, and (**c**) tetragonal phases of ZrO_2_; (**d**) model of 2 × 2 × 2 supercell of monoclinic ZrO_2_ doped with Y_2_O_3_; (**e**,**f**) yttrium substitution sites in the surface matrix, and (**g**) a box with a 35 Å vacuum containing water molecules from the surface of the ZrO_2_ substrate.

**Figure 2 nanomaterials-13-02657-f002:**
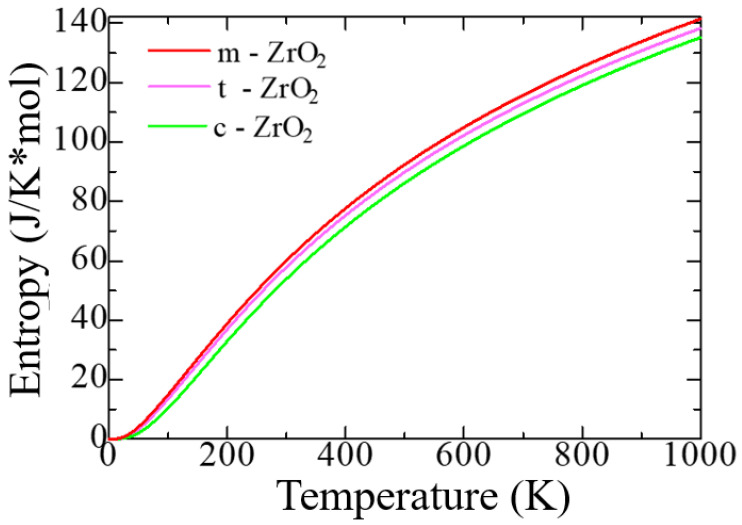
Entropy as a function of absolute temperature per unit cell.

**Figure 3 nanomaterials-13-02657-f003:**
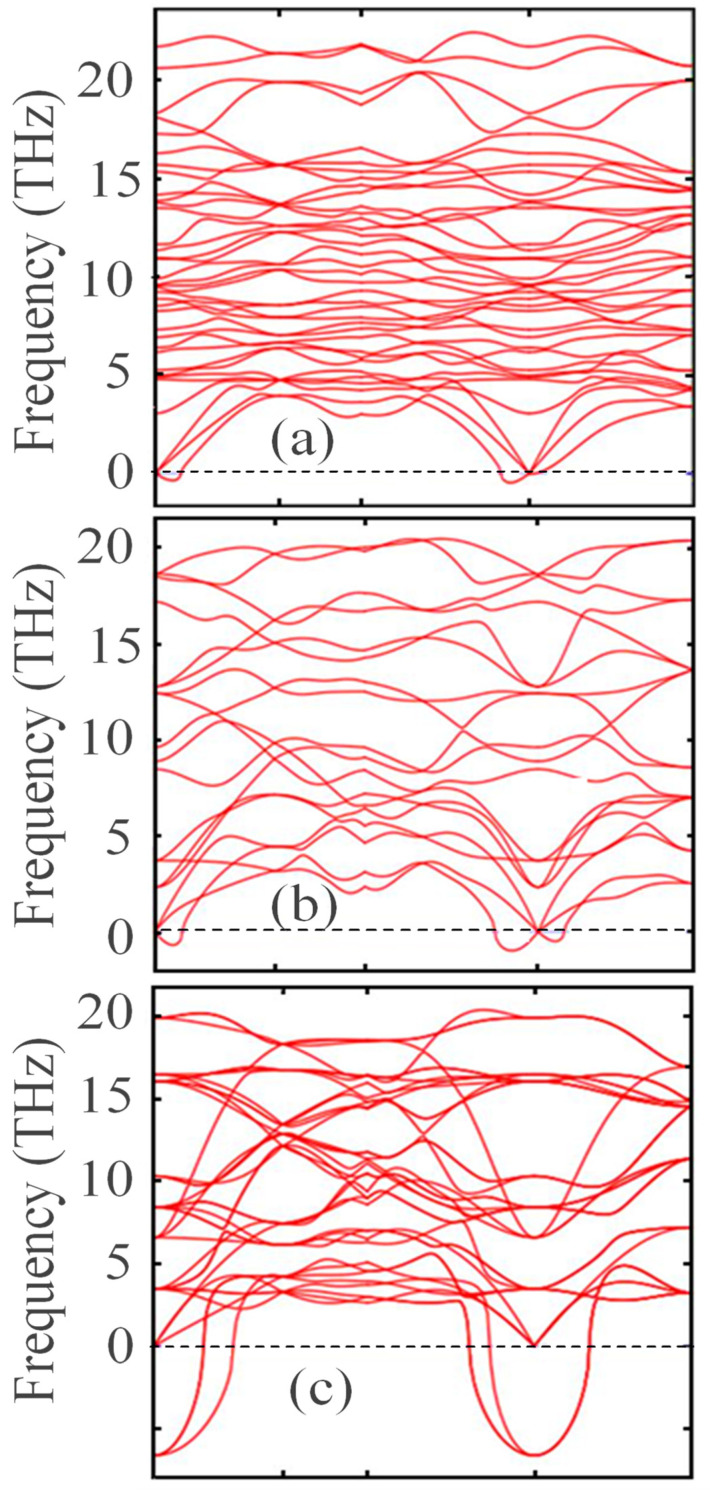
Phonon dispersion relations of (**a**) monoclinic, (**b**) tetragonal, and (**c**) cubic ZrO_2_ at 0 K.

**Figure 4 nanomaterials-13-02657-f004:**
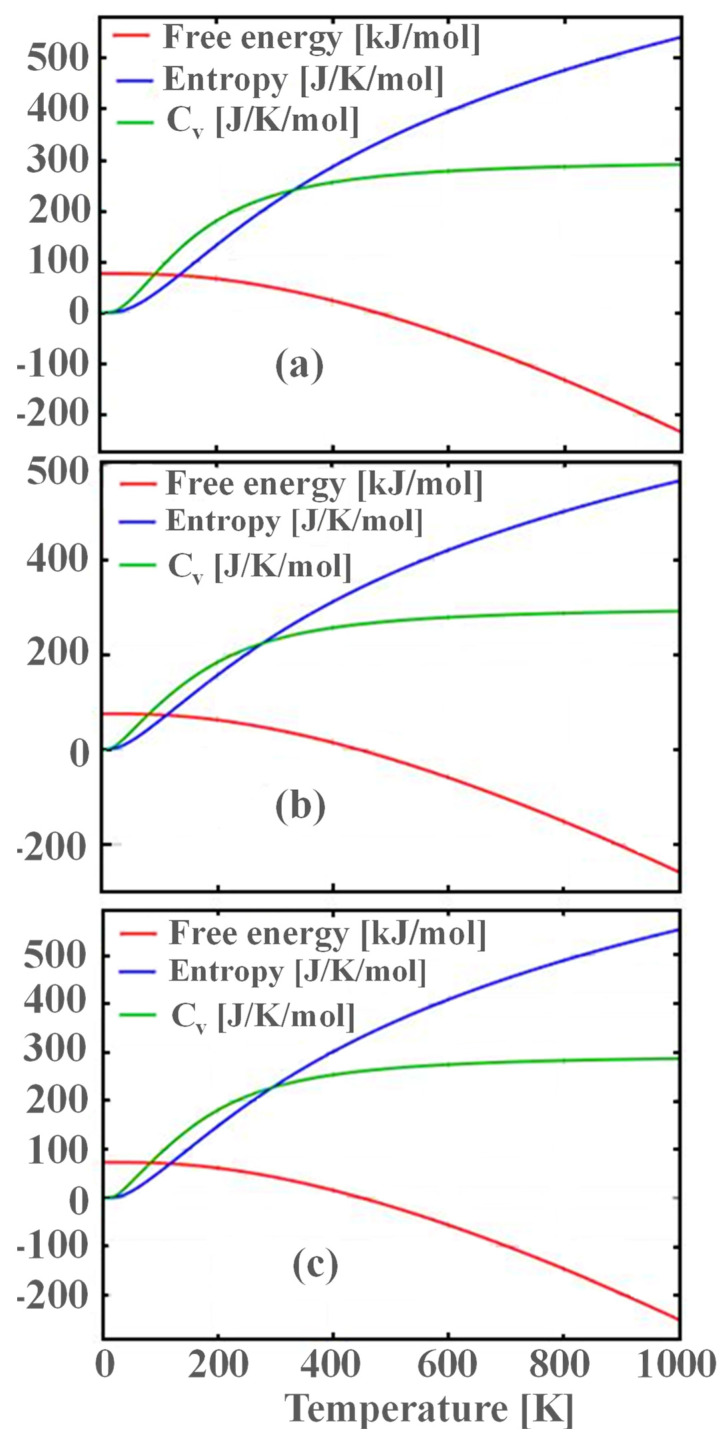
Temperature dependence of free energy, entropy, and heat capacity of a 12-atom supercell for m-ZrO_2_ (**a**), t-ZrO_2_ (**b**), and с-ZrO_2_ (**c**).

**Figure 5 nanomaterials-13-02657-f005:**
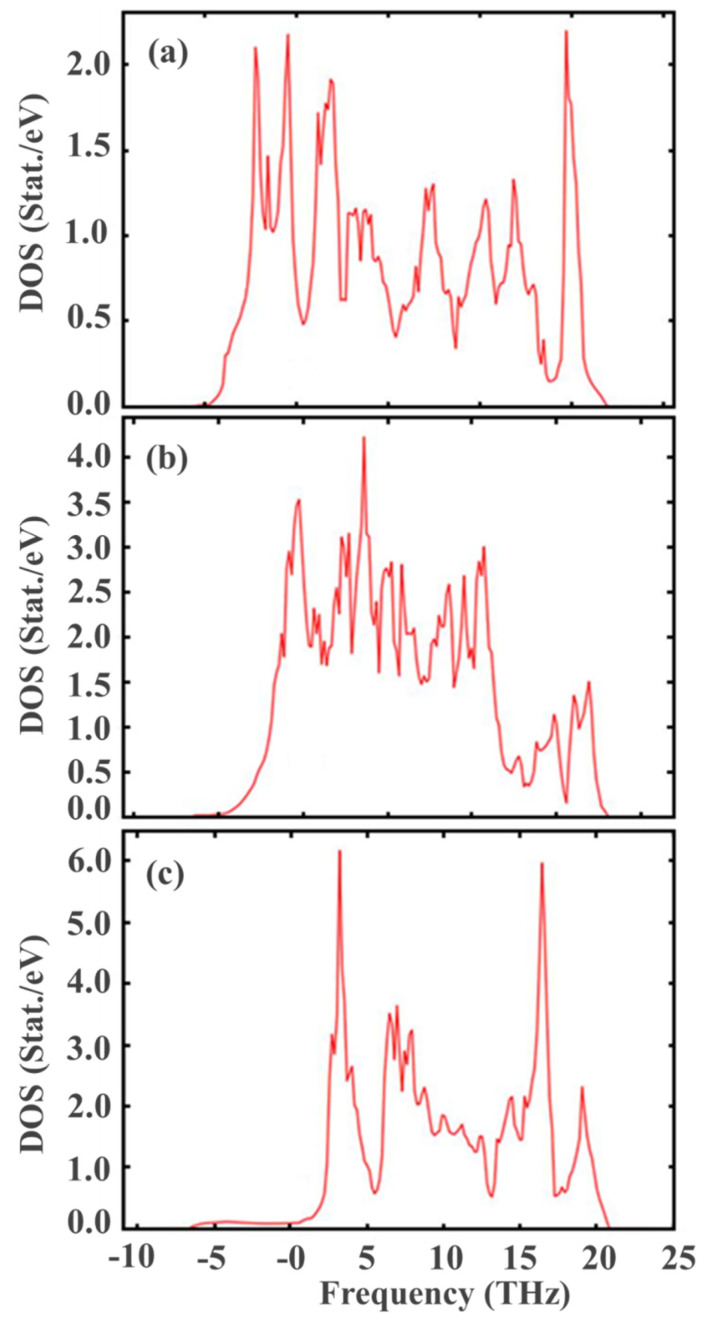
Phonon density of state for m-ZrO_2_ (**a**), t-ZrO_2_ (**b**), and с-ZrO_2_ (**c**).

**Figure 6 nanomaterials-13-02657-f006:**
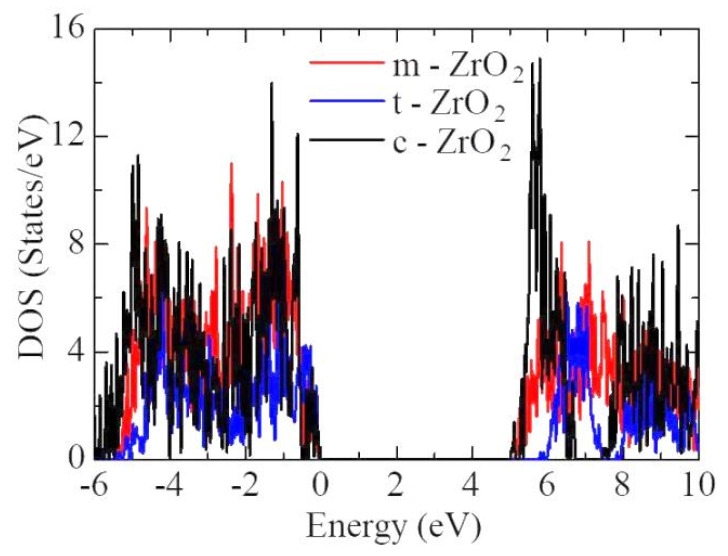
The total density of electronic states (TDOS) of monoclinic, tetragonal, and cubic ZrO_2_.

**Figure 7 nanomaterials-13-02657-f007:**
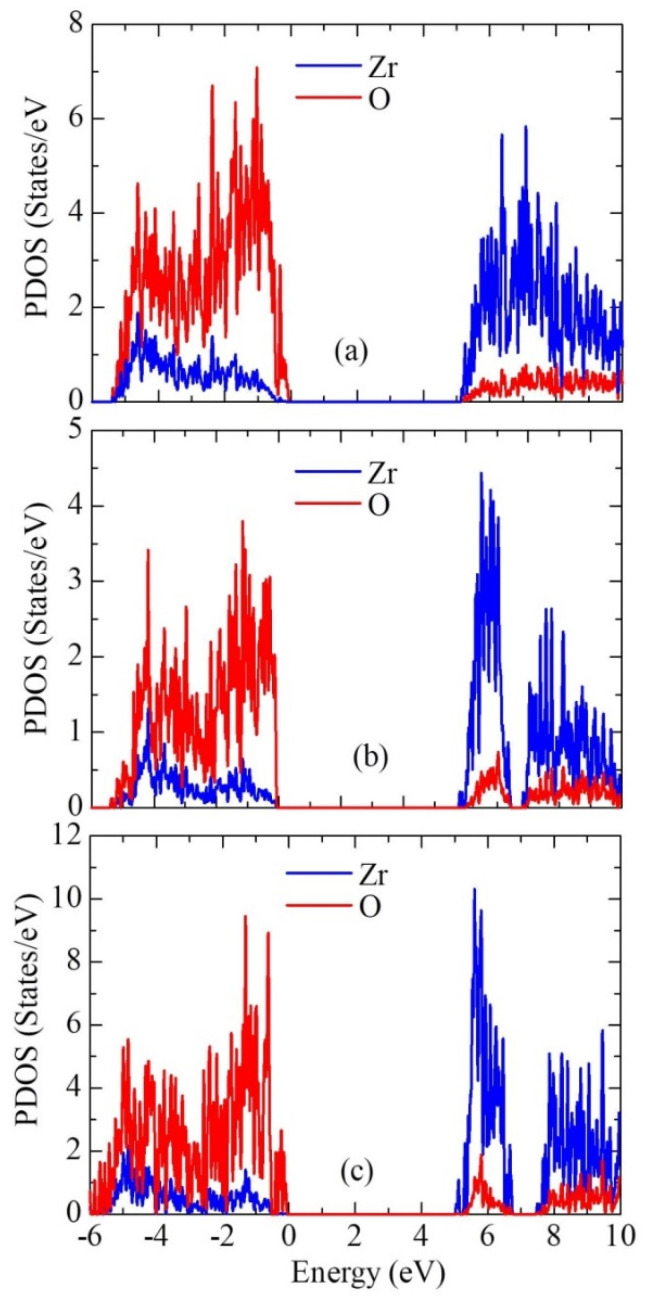
Partial density of electronic states (PDOS) for (**a**) monoclinic, (**b**) tetragonal, and (**c**) cubic ZrO_2_.

**Figure 8 nanomaterials-13-02657-f008:**
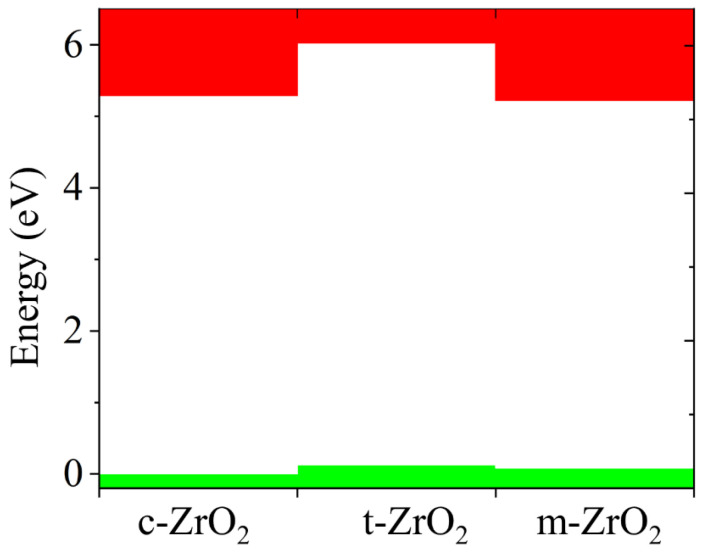
Conduction (red) and valence (green) band changes for c-ZrO_2_, t-ZrO_2_, and m-ZrO_2._ The position of the Fermi level corresponds to the maximum of the valence band at each of the sites.

**Figure 9 nanomaterials-13-02657-f009:**
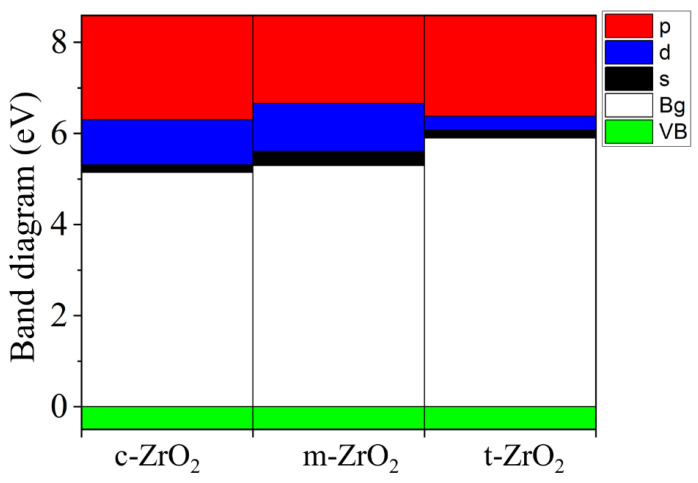
Composite PDOS diagram showing the main contributions of the s-, p-, and d-orbitals to the states that form the conduction band bottom for c-ZrO_2_, t-ZrO_2_, and m-ZrO_2_. Top valence band (green) scaled to zero.

**Figure 10 nanomaterials-13-02657-f010:**
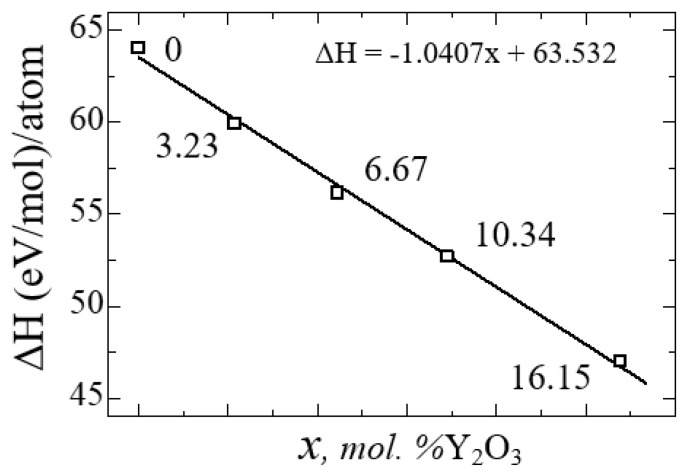
Enthalpy formation energy for YSZ as a function of Y_2_O_3_ concentration.

**Figure 11 nanomaterials-13-02657-f011:**
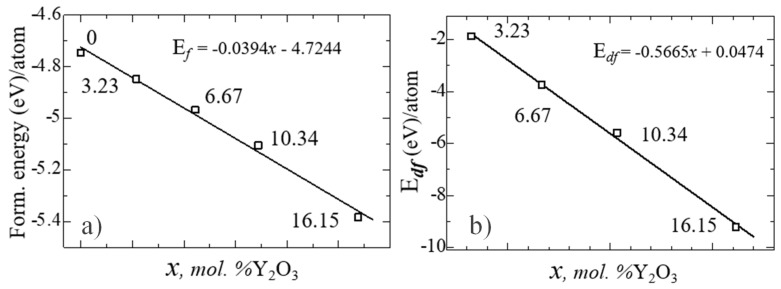
Formation energy (**a**) and formation energy of an oxygen vacancy (**b**) for YSZ as a function of Y_2_O_3_ concentration.

**Figure 12 nanomaterials-13-02657-f012:**
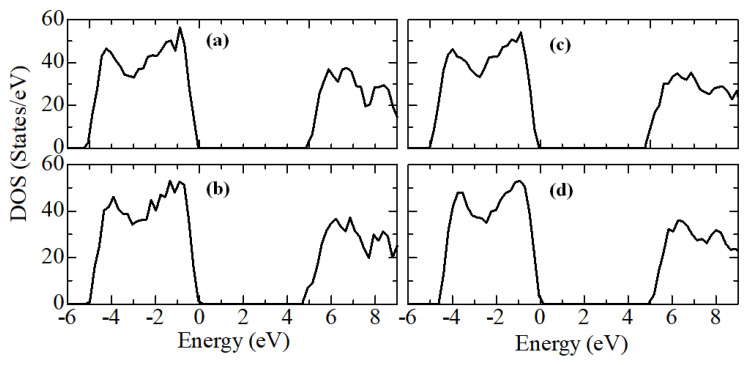
Total density of electronic states (TDOS) for ZrO_2_ doped with 3.23 mol. %Y_2_O_3_ (**a**), 6.67 mol. %Y_2_O_3_ (**b**), 10.34 mol. %Y_2_O_3_ (**c**), and 16.15 mol. %Y_2_O_3_ (**d**).

**Figure 13 nanomaterials-13-02657-f013:**
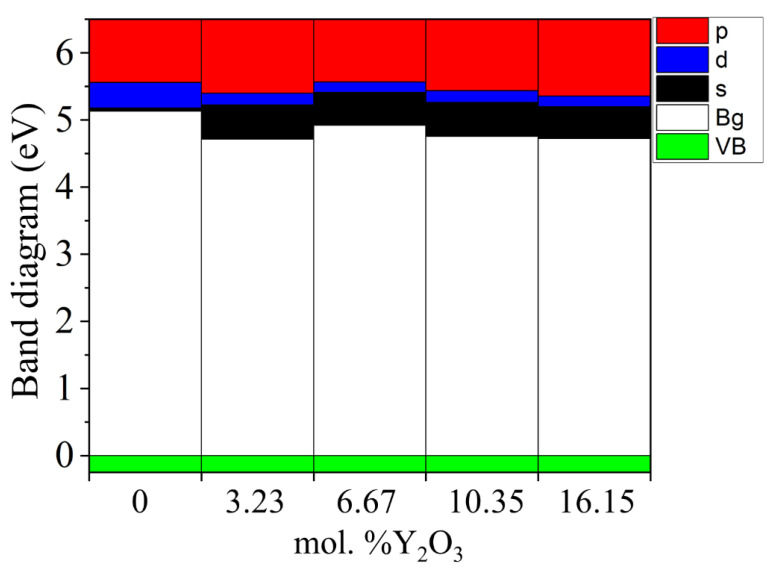
Composite PDOS diagram showing the main contributions of the s-, p-, and d-orbitals to states forming the conduction band bottom for ZrO_2_ doped with 3.23, 6.67, 10.34, and 16.15 mol. %Y_2_O_3_. Top valence band (green), scaled to zero.

**Figure 14 nanomaterials-13-02657-f014:**
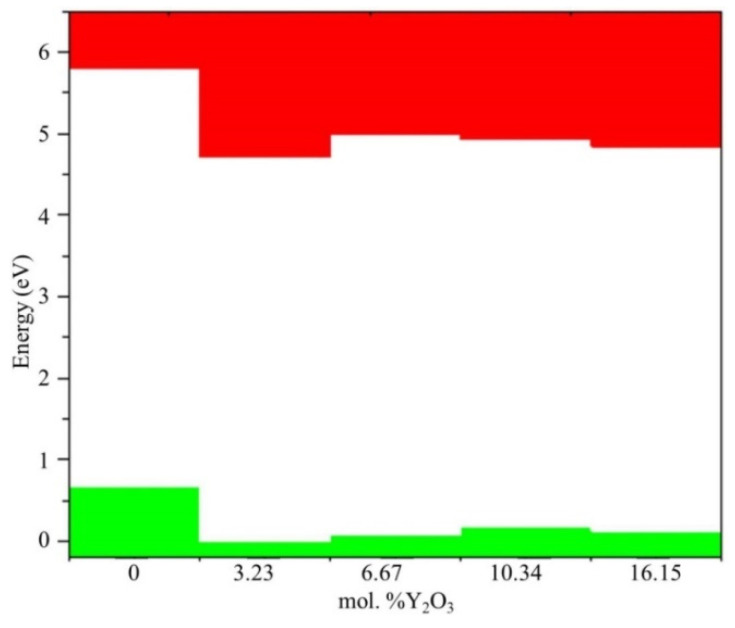
Conduction (red) and valence (green) band changes for ZrO_2_ doped with 3.23, 6.67, 10.34, and 16.15 mol. %Y_2_O_3_. The position of the Fermi level corresponds to the maximum valence band in each section.

**Figure 15 nanomaterials-13-02657-f015:**
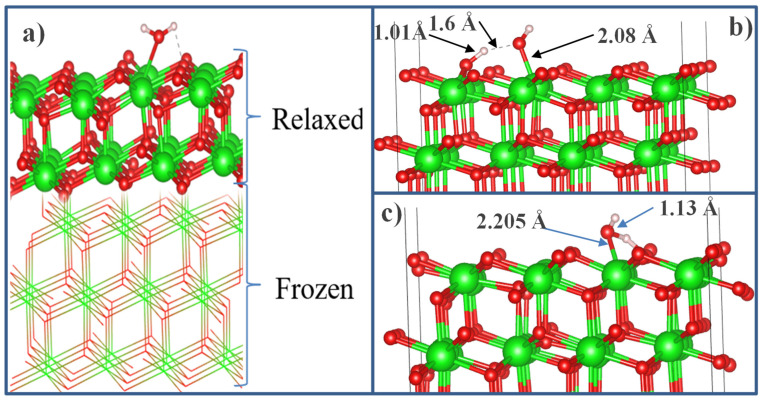
Configuration of water molecule adsorption on the surface of t-ZrO_2_ (101): (**a**) model of a lamellar t-ZrO_2_ (101) cell with the initial configuration of water on its surface, (**b**) dissociative adsorption in a side view, (**c**) model of molecular physisorption of water on the surface of t-ZrO_2_ in a side view.

**Figure 16 nanomaterials-13-02657-f016:**
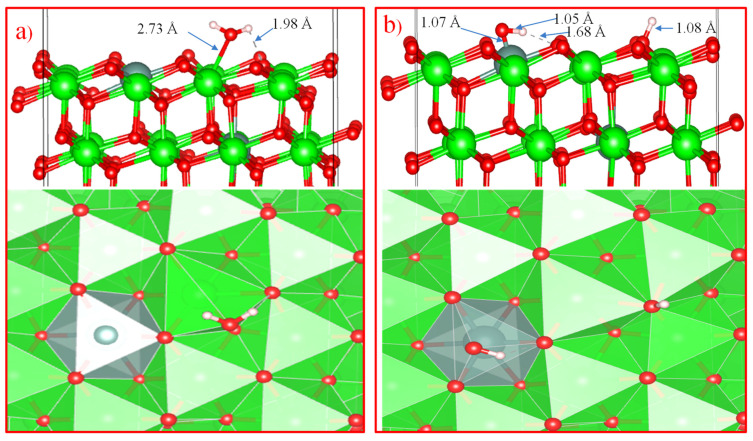
Molecular (**a**) and dissociated adsorption of water to form surface hydroxyls (**b**) in the H_2_O–YSZ (101) model.

**Figure 17 nanomaterials-13-02657-f017:**
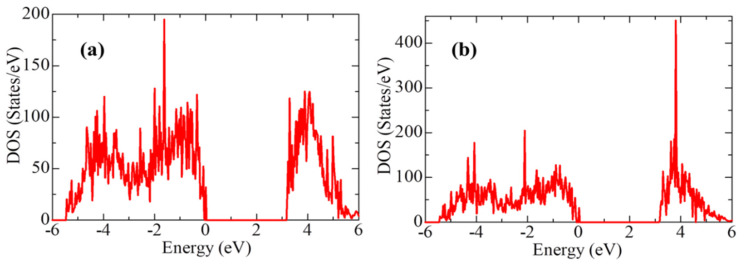
Total density of states (TDOS) calculated by the GGA for H_2_O molecules adsorbed on: (**a**) t-ZrO_2_ (101) and (**b**) t-YSZ (101).

**Figure 18 nanomaterials-13-02657-f018:**
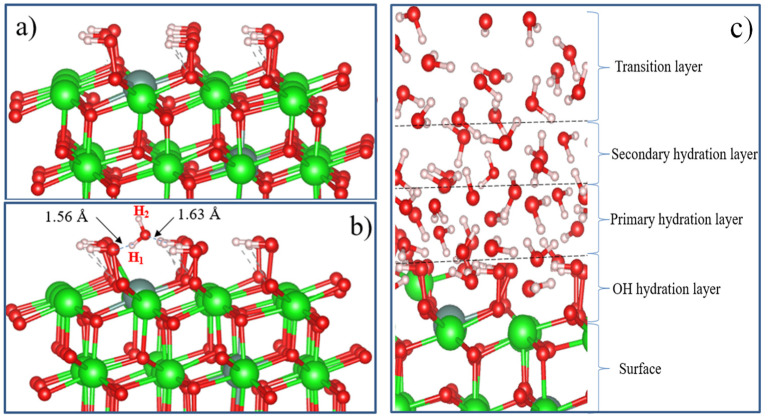
Relaxation configurations: (**a**) fully hydroxylated t-YSZ (101); (**b**) single water adsorption on a fully hydroxylated t-YSZ (101) surface; and (**c**) t-YSZ (101) surface hydration model.

**Table 1 nanomaterials-13-02657-t001:** Relaxation parameters of the ZrO_2_ phase. The calculation results are compared to experimental and previous theoretical results.

	Lattice Constants	This Work		Another Calc. [55]	Exp.
		GGA	SCAN		
m-ZrO_2_ [P2_1/c]	a (Å)	5.191	5.115	5.090	5.0950 [56]
	b (Å)	5.245	5.239	5.187	5.2116 [56]
	c (Å)	5.202	5.304	5.243	5.3173 [56]
	*β*◦	99.639	99.110	99.432	99.230 [56]
	V (Å^3^)	144.410	139.400	137.76	140.88 [56]
	E − E_m_ (eV/ZrO_2_)	0	0	0	0
t-ZrO_2_ [P4_2/nmc]	a = b (Å)	3.593	3.622	-	3.64 [57]
	с (Å)	5.193	5.275	-	5.27 [57]
	c/a	1.445	1.456	-	1.45 [57]
	V (Å^3^)	67.05	69.214	-	69.83 [57]
	dz	0.012	0.013	1.011	0.046 [58]
	E − E_m_ (eV/ZrO_2_)	0.4257	0.4257	0.048	0.065 [59]
c-ZrO_2_ [Fm-3m]	a = b = c (Å)	5.075	5.12	5.031	5.129 [60]
	V(Å^3^)	130.709	134.06	127.36	134.9 [60]
	E − E_m_ (eV/ZrO_2_)	0.833	0.833	0.087	0.140 [61]

**Table 2 nanomaterials-13-02657-t002:** GGA-calculated total electron energies of ZrO_2_ unit cells.

System	Energy	ΔE
m-ZrO_2_	−28.7947	0
t-ZrO_2_	−28.6885	0.106
c-ZrO_2_	−28.5865	0.208

**Table 3 nanomaterials-13-02657-t003:** Calculated and experimental band gap of ZrO_2_ in eV.

System	This Work			Experiment [62]
	GGA	SCAN	HSE06	VUV
m-ZrO_2_	3.9	3.8	5.288	5.78
t-ZrO_2_	4.42	4.37	5.898	5.83
c-ZrO_2_	4.03	3.93	5.140	6.10

**Table 4 nanomaterials-13-02657-t004:** The number of Zr, Y, and O ions for various mol. %Y_2_O_3_, taking into account the oxygen vacancies.

mol. %Y_2_O_3_	Zr	Y	O	O Vacancy	System
0	32	0	64	0	Zr_32_O_64_
3.23	30	2	63	1	Zr_30_Y_2_O_63_
6.67	28	4	62	2	Zr_28_Y_4_O_62_
10.35	26	6	61	3	Zr_26_Y_6_O_61_
16.15	22	10	59	5	Zr_22_Y_10_O_59_

**Table 5 nanomaterials-13-02657-t005:** Lattice parameters of 2 × 2 × 2 supercells of ZrO_2_ and YSZ at various Y_2_O_3_ concentrations.

System	Lattice Parameters						Structure
	a (Å)	b (Å)	c (Å)	α (^◦^)	β (^◦^)	γ (^◦^)	
0	10.382	10.491	10.757	90	99.64	90.00	m-YSZ
3.23 mol. %Y_2_O_3_	10.274	10.524	10.536	90.21	98.84	89.94	m-YSZ
6.67 mol. %Y_2_O_3_	10.512	10.544	10.603	89.90	90.12	89.62	t-YSZ
10.35 mol. %Y_2_O_3_	10.529	10.541	10.546	89.98	90.09	90.08	t-YSZ
16.15 mol. %Y_2_O_3_	10.540	10.541	10.543	90.08	90.00	90.02	c-YSZ

**Table 6 nanomaterials-13-02657-t006:** GGA-calculated values of enthalpy (ΔН) and energy of formation ($E_{f}$) for ZrO_2_ and YSZ. Oxygen vacancy formation energy ($E_{df}$) for YSZ.

System	ΔН	$E_{f}$	$E_{df}$
0	64.02917222	−4.747216667	0
3.23 mol. %Y_2_O_3_	59.91124404	−4.848422632	−1.874577368
6.67 mol. %Y_2_O_3_	56.13271879	−4.967857447	−3.739875532
10.35 mol. %Y_2_O_3_	52.7041267	−5.106527419	−5.596013441
16.15 mol. %Y_2_O_3_	47.00229139	−5.384704945	−9.220196154

**Table 7 nanomaterials-13-02657-t007:** Calculated values of surface energies (σ) for the main types of ZrO_2_ plate.

Phase	Miller Indices	Surface Energy [σ, 10^19^ eV/m^2^]
m-ZrO_2_	(001)	1.54
	(010)	1.16
	(110)	1.10
	(101)	1.23
	(011)	1.08
	(111)	0.81
t-ZrO_2_	(001)	0.98
	(010)	0.95
	(101)	0.78
	(100)	1.01
	(111)	0.79
c-ZrO_2_	(100)	1.51
	(110)	1.34
	(111)	1.12

**Table 8 nanomaterials-13-02657-t008:** Adsorption energies (E_ads_) and structural characteristics of t-ZrO_2_ (101) and t-YSZ (101) with adsorbed water.

	t-ZrO_2_ (101)	t-YSZ (101)
E_ads_ (H_2_O), eV	-	−1.84
E_ads_ (H+OH), eV	−1.22	−1.23
Dist O(H_2_O)–surf, Å	2.08	2.73
Dist O(H_2_O)–H1(H_2_O), Å	0.97	0.96
Dist O(H_2_O)–H2(H_2_O), Å	1.13	0.97
H–O–H bond angle, (°)	111.3	105.54

## Data Availability

The data presented in this study are available on request from the corresponding authors.

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
