# Peer review of "A Detailed Comparative Analysis of the Structural Stability and Electron-Phonon Properties of ZrO2: Mechanisms of Water Adsorption on t-ZrO2 (101) and t-YSZ (101) Surfaces"

_nanomaterials, 2023, doi:10.3390/nano13192657_

Round 1

Reviewer 1 Report

The stability, electronic properties, and dispersion of phonons in the three phases of the ZrO2 phases were investigated to investigate the mechanism of water adsorption on t-ZrO2 (101) and t-YSZ (101) surfaces. The topic of this paper is nice and this paper reports interesting results. However, the author needs to address the following comments before they can be accepted for publication.

About the writing:

1- Proofread the manuscript and correct grammatical errors. For example, define all the acronym (such as GGA, DFT, PVW, and so on) in your manuscript. This is obvious for a sector of scientists but not for everyone who might be interested in ZrO2. Please use the standard three-line table.
2- Please use uniform standards for drawing all figures in your manuscript.
3- Provide clear figures to make data interpretation easier, especially figures 8,17,18, and 20.
4- Use italics for variables. Please note the difference between the use of italics and normal.
5- The references list doesn't contain enough references for studies conducted in 2023 (just 2 references). Add some review references to your references list in the introduction section.

So on……

About the content:

1-     Please explain in detail why a 4x4x4 k-point grid with the Monkhrost-Pack scheme is optimal for the geometric relaxation of ZrO2. And why choose cubic phases to carry out the k-point convergence test, generally the cubic phase is not the main phase in ZrO2 and YSZ powders.

2-     The relationship between electron-phonon properties in the mechanism of water adsorption is not closely linked, please explain in detail. 

Considering the amount of mistakes and poor research paper writing skill present in this manuscript, a further check carried out by a professional English language center or professional research paper writing institution is suggested.

Author Response

Response to Reviewer 1 Comments

I express my deep gratitude to the respected reviewer for the time allocated for reviewing this article, valuable advice and recommendations.

Point 1: Proofread the manuscript and correct grammatical errors. For example, define all the acronym (such as GGA, DFT, PVW, and so on) in your manuscript. This is obvious for a sector of scientists but not for everyone who might be interested in ZrO2. Please use the standard three-line table.

Response 1: All reductions have been identified and recommendations taken into account. Thank you!

Point 2: Please use uniform standards for drawing all figures in your manuscript.

Response 2: The recommendation was noted and corrected. Thank you!

Point 3: Provide clear figures to make data interpretation easier, especially figures 8,17,18, and 20.

Response 3: The recommendation was noted and corrected. Thank you!

Point 4: Use italics for variables. Please note the difference between the use of italics and normal.

Response 4: All required fixes are entered, variables are marked in italics.  Thank you!

Point 5: The references list doesn't contain enough references for studies conducted in 2023 (just 2 references). Add some review references to your references list in the introduction section.

Response 5: Added 16 references to articles published in 2022-2023.  Thank you!

Point 6: Please explain in detail why a 4x4x4 k-point grid with the Monkhrost-Pack scheme is optimal for the geometric relaxation of ZrO2. And why choose cubic phases to carry out the k-point convergence test, generally the cubic phase is not the main phase in ZrO2 and YSZ powders. Response 6: Dear reviewer, thanks for the good question!

The fact is that the cubic phase is the most symmetrical among all other phases of zirconia, and with a symmetrical structure containing a small number of atoms, it is easier to carry out convergence tests and takes little time. On the other hand, with a uniform number of k-points in the Monkhrost-Pack scheme, it is easier to perform convergence tests than with different numbers. Therefore, it was easier to perform ENCUT and k-point tests for the cubic phase. We chose the 4x4x4 scheme, since starting from this value the energy remains unchanged, and a further increase in the k-point only increases the calculation time.  Thank you!

Point 7: The relationship between electron-phonon properties in the mechanism of water adsorption is not closely linked, please explain in detail.

Response 7: Dear reviewer, you noted correctly. This has nothing to do with the adsorption of a water molecule on a surface. Phononic and thermodynamic properties were calculated to find and accurately evaluate stable phases and surface types for water adsorption.  Thank you!

Reviewer 2 Report

Subject: Review of Manuscript Submission nanomaterials-2559650

I have carefully reviewed the manuscript titled "Detailed Comparative Analysis of Structural Stability and Electron-Phonon Properties of ZrO2: Mechanism of Water Adsorption on t-ZrO2 (101) and t-YSZ (101) Surfaces" submitted to Nanomaterials for consideration. Dilshod et al develop the demonstration of structural stability, electronic properties, and phonon dispersion of the cubic, tetragonal, and monoclinic phases of ZrO2, as well illustrate the mechanism of water adsorption on the surface of t-ZrO2 (101) and t-YSZ (101). I think this article is meaningful and can provide both a basis and directions for future research. It can be accepted after revision.

1. As we all know, GGA method has wrong self-interaction. Due to the undeserved interaction between electrons and themselves, electrons are more delocalized. The GGA's calculated energy should always be more positive than the true value. When referring to the bandgaps and DOS of oxide, the HSE06 package is more commonly adopted, which can realize a superior accurate result than GGA. The authors should explain why the HSE06 package was not applied.

2.  In GGA method, the on-site repulsion U is as usual introduced in GGA+U for correcting SIE in GGA, which is suitable for the strong electron correlation system. Thus, why didn't U be utilized to correct the system?

3. The schematic in the introduction section is strange. Just citing the relevant literature is enough. Moreover, the phase transition temperature is not directly related to the article, so it should be deleted.

4. Figure. 2 and Figure. 3 are both redundant, which can only provide partial information of calculation. I suggest deleting them, and just illustrating them in the experimental section.

5. Figure. 5, Figure. 6, and Figure. 9 are significantly compressed, in which the labels and serial numbers, as well as the illustrations, are wrong. Moreover, the location of serial numbers in each Figure should be fixed, for example, Figure. 5 (a) (b) (c), Figure. 6 (a) (b) (c), Figure. 7 (a) (b) (c).

6. The TDOS for m-ZrO2, t-ZrO2, and с-ZrO2 should be put together in a single figure, for the convenience of the readers to compare the differences between the different types. And the PDOS of different atoms in different ZrO2 should be analyzed.

7. Need to improve the English language though out the article.

Need to improve the English language though out the article.

Author Response

Response to Reviewer 1 Comments

I express my deep gratitude to the respected reviewer for the time allocated for reviewing this article, valuable advice and recommendations.

Point 1: As we all know, GGA method has wrong self-interaction. Due to the undeserved interaction between electrons and themselves, electrons are more delocalized. The GGA's calculated energy should always be more positive than the true value. When referring to the bandgaps and DOS of oxide, the HSE06 package is more commonly adopted, which can realize a superior accurate result than GGA. The authors should explain why the HSE06 package was not applied.

Response 1: Dear reviewer, sorry for our mistakes. All kinds of electronic property calculations were performed using the HSE06 hybrid potential, however this was not noted in the text. Only in the sentence " Next, for ZrO2 structures relaxed using the HSE06 functional" in the text was erroneously written SCAN. This bug has been fixed in this version. . Thank you!

Point 2: In GGA method, the on-site repulsion U is as usual introduced in GGA+U for correcting SIE in GGA, which is suitable for the strong electron correlation system. Thus, why didn't U be utilized to correct the system?

Response 2: The Leonard Jones potential was used to account for the long-range forces of attraction and repulsion. The only problem is that the authors do not yet have experience in calculations using the Hubbard parameter (U). However, in future research, we will definitely try to take into account your valuable recommendations.

Point 3: The schematic in the introduction section is strange. Just citing the relevant literature is enough. Moreover, the phase transition temperature is not directly related to the article, so it should be deleted.

Response 3: The recommendation was noted and corrected. Thank you!

Point 4: Figure. 2 and Figure. 3 are both redundant, which can only provide partial information of calculation. I suggest deleting them, and just illustrating them in the experimental section.

Response 4: The recommendation was noted and corrected. Figures 2 and 3 have been removed. Thank you!

Point 5: Figure. 5, Figure. 6, and Figure. 9 are significantly compressed, in which the labels and serial numbers, as well as the illustrations, are wrong. Moreover, the location of serial numbers in each Figure should be fixed, for example, Figure. 5 (a) (b) (c), Figure. 6 (a) (b) (c), Figure. 7 (a) (b) (c).

Response 5: The recommendation was noted and corrected. Thank you!

Point 6: The TDOS for m-ZrO2, t-ZrO2, and с-ZrO2 should be put together in a single figure, for the convenience of the readers to compare the differences between the different types. And the PDOS of different atoms in different ZrO2 should be analyzed.

Response 6: The recommendation was noted and corrected. Thank you!

Point 7: Need to improve the English language though out the article.

Response 7: Dear reviewer, thank you for your advice. Indeed, it was necessary to correct the English language in the article. I am sending a request to the MDPI editing service to correct the language in this article.  Thank you!

If there are other tips and comments, the authors will gladly correct the article. Thank you for your review.

Reviewer 3 Report

This work deals with Detailed Comparative Analysis of Structural Stability and Electron-Phonon Properties of ZrO2: Mechanism of Water Adsorption on t-ZrO2 (101) and t-YSZ (101) Surfaces. The authors’ work is very interesting in the related fields. However, for next the step, some parts should be improved and questions should be solved.

1. Firstly, the resolution of figures is very low. I recommend you would improve the figures.

2. Could you explain the different simulation modeling and experimental data?

3. In Water Adsorption, what is the state of the surface of ZrO2? hydrophobic and hydrophilic?

4. In Figure 2, after 2x2x2 MP k-point mesh, it seems like all energy is the same. I recommend you would zoom in to show parts 2x2x2 to 4x4x4.

I recommend you would check the typos and errors in the manuscript.

Author Response

Response to Reviewer 1 Comments

I express my deep gratitude to the respected reviewer for the time allocated for reviewing this article, valuable advice and recommendations.

Point 1: Firstly, the resolution of figures is very low. I recommend you would improve the figures.

Response 1: The recommendation was noted and corrected. Thank you!

Point 2: Could you explain the different simulation modeling and experimental data?

Response 2: Dear reviewer, thank you for your question. Indeed, it was possible to explain in detail the various simulation models and experimental data, however, there are few detailed results in this direction in the literature and there is no data to compare all the results with experiment. Only a few results have been compared with their experimental counterparts. Thank you!

Point 3: In Water Adsorption, what is the state of the surface of ZrO2? hydrophobic and hydrophilic?

Response 3: Thanks for the question. The surface of zirconia is hydrophilic and well suited for water adsorption. Thank you!

Point 4 In Figure 2, after 2x2x2 MP k-point mesh, it seems like all energy is the same. I recommend you would zoom in to show parts 2x2x2 to 4x4x4.

Response 4: Dear reviewer, I apologize to you for not being able to fulfill your recommendation, since Figure 2 was removed at the request of two previous reviewers, since one of them considered it superfluous. Thanks for your review.

Point 7: I recommend you would check the typos and errors in the manuscript.

Response 7: Dear reviewer, thank you for your advice. Indeed, it was necessary to correct the English language in the article. I am sending a request to the MDPI editing service to correct the language in this article.  Thank you!

I hope you are satisfied with our answers. If there are other tips and comments, the authors will gladly correct the article. Thank you for your review.

Round 2

Reviewer 2 Report

I am satisfied with the revision. And I recommend this manuscript can be published in its present form.

Author Response

Dear reviewer, thank you for your consent to publish this article.
